

**Variability of Black Carbon mass concentration in surface snow at Svalbard**
Michele Bertò[1#], David Cappelletti[2,7], Elena Barbaro[1,3], Cristiano Varin[1], Jean-Charles Gallet[4], Krzysztof
Markowicz[5], Anna Rozwadowska[6], Mauro Mazzola[7], Stefano Crocchianti[2], Luisa Poto[1,3], Paolo Laj[8],
Carlo Barbante[1,3] and Andrea Spolaor[1,2*].
[1]Ca' Foscari University of Venice, Dept. Environmental Sciences, Informatics and Statistics, via Torino,
155 - 30172 Venice-Mestre, Italy;
[2]Università degli Studi di Perugia, Dipartimento di Chimica, Biologia e Biotecnologie, Perugia, Italy;
[3]CNR-ISP, Institute of Polar Science – National Research Council –via Torino, 155 - 30172 Venice-
Mestre, Italy;
[4]Norwegian Polar Institute, Tromsø, Norway.
[5]University of Warsaw, Institute of Geophysics, Warsaw, Poland;
[6]Institute of Oceanology, Polish Academy of Sciences, Sopot, Poland;
[7]CNR-ISP, Institute of Polar Science – National Research Council – Via Gobetti 101, Bologna;
[8]Univ. Grenoble-Alpes, CNRS, IRD, Grenoble-INP, IGE, 38000 Grenoble, France
[#]Now at Laboratory of Atmospheric Chemistry, Paul Scherrer Institute, 5232 Villigen PSI, Switzerland
Correspond to: Andrea Spolaor, andrea.spolaor@cnr.it; Michele Bertò, michele.berto@gmail.it
**Abstract**
Black Carbon (BC) is a significant forcing agent in the Arctic, but substantial uncertainty remains
to quantify its climate effects due to the complexity of the different mechanisms involved, in particular
related to processes in the snow-pack after deposition. In this study, we provide detailed and unique
information on the evolution and variability of BC content in the upper surface snow layer during the
spring period in Svalbard (Ny-Ålesund). Two different snow-sampling strategies were adopted during
spring 2014 and 2015, providing the *refractory* BC (rBC) mass concentration variability on a
seasonal/daily and daily/hourly time scales. The present work aims to identify which atmospheric
variables could interact and modify the mass concentration of BC in the upper snowpack, the snow layer
which BC particles affects the snow albedo. Atmospheric, meteorological, and snow-related physico-
chemical parameters were considered in a multiple linear regression model to identify the factors that
could explain the variations of BC mass concentrations during the observation period. Precipitation



events were the main drivers of the BC variability. Snow metamorphism and activation of local sources
during the snow melting periods appeared to play a non-negligible role (wind resuspension in specific
Arctic areas where coal mines were present). The statistical analysis suggests that the BC content in the
snow is decoupled from the atmospheric BC load.

## 1. Introduction

In the last two decades, the Arctic region has been exposed to dramatic changes in terms of

atmospheric temperature rise, sea ice decrease, and increase of air mass transport from lower latitudes
bringing warmer and humid air masses containing pollutants and anthropogenic derived compounds (Law
and Stohl, 2007; Comiso et al., 2008; Screen and Simmonds, 2010; Eckhardt et al., 2013; Schmale et al.,
2018; Maturilli et al., 2019). Long-range transport and local emissions of combustion generating aerosols
like black carbon (BC) can influence the radiative budget of the Arctic atmosphere, especially  the
impacts of atmospheric aging on the mixing state of BC particles (Eleftheriadis et al., 2009; Bond et al.,
2013; Zanatta et al., 2018). When deposited over snow, numerous aerosol species directly increase the
quantity of solar radiation absorbed by the snowpack, thus favouring snow aging processes and the
decrease of the snow albedo (Hansen and Nazarenko, 2004; Flanner et al., 2007; Hadley and Kirchstetter,
2012; Skiles et al., 2018; Skiles and Painter, 2019).

Among these light-absorbing aerosols, *black carbon* (BC) particles are the most effective in

absorbing the visible and near infrared solar radiation. These primarily emitted, insoluble, refractory and
carbonaceous particles originate from natural and anthropogenic sources such as open fires or diesel
engine exhausts. Currently, the anthropogenic emissions are higher compared to the natural ones
(Moosmüller et al., 2009; Bond et al., 2013). In 2000 the energy production sector (including fossil fuels
and solid residential fuels combustion) generated approximately 59% of the total global BC emissions
while the remaining came from biomass burning (Bond et al., 2013). BC particles are characterized by a
mass size distribution peaking around 100-250 nm (or mass equivalent diameter), e.g. 240 nm in the
Svalbard area in spring  (Bond et al., 2013; Laborde et al., 2013; Zanatta et al., 2016; Motos et al., 2019).
The impact of BC particles absorbing the incoming solar radiation has indeed a non-negligible role in the
Arctic region, which is already threatened by a two-fold temperature increase compared to the mid-
latitude areas, the so-called "Arctic Amplification" (Bond et al., 2013; Cohen et al., 2014; Serreze and
Barry, 2011). BC has an atmospheric lifetime of about seven days and has been directly targeted in
important international mitigation agreements (AMAP, 2015). Theoretical and experimental results
showed that the cryosphere is affected both by the BC-induced warming of the atmosphere and by direct
and indirect BC effects on the snow once deposited over it (Flanner, 2013),



Atmospheric BC measurements in the Arctic regions are still rare, despite an extraordinary effort
done by the international scientific community to evaluate the sources, transport paths, concentration, and
climate impact  (Eleftheriadis et al., 2009; Pedersen et al., 2015; Ferrero et al., 2016; Ruppel et al., 2017;
Osmont et al., 2018; Zanatta et al., 2018; Laj et al., 2020). BC mass concentrations can be measured
directly by using incandescent or thermal techniques and indirectly, by absorption measurements using an
appropriate mass absorption cross-section (Petzold, 2013). Various terms such as refractory black carbon
(rBC) for incandescent measurements, elemental carbon (EC) using thermal techniques, or equivalent
black carbon (eBC) based on optical technique are used. Forsström et al. (2009) reported measurements
performed in Arctic snow in the past and new measurements of EC in snow surface using filters and a
thermo-optical method. The geographical and seasonal eBC variability was investigated in the Arctic
region by Doherty et al. (2010). Other BC measurement in snow samples from the Arctic region can be
found in Aamaas et al. (2011), Forsström et al. (2013), Pedersen et al. (2015), Gogoi et al. (2016), Khan
et al. (2017) and Mori et al. (2019). Intercomparison of different techniques agree within a factor of 2
uncertainty at Alert (Sharma et al., 2017), Ny-Ålesund, and Barrow (Sinha et al., 2017).
A complex combination of processes are involved in the BC particles transfer from the
atmosphere to the surface snow. Via a modelling approach, Liu et al. (2011) found that approximately
50% of BC's total burden in the Arctic atmosphere is removed through wet deposition-related processes.
Yasunari et al. (2013) estimated the intensity of BC dry deposition on the Himalayan glaciers; they found
that the surface roughness and the surface wind speed are critical parameters in order to retrieve realistic
results. Emerson et al. (2018) empirically evaluated the in situ rBC deposition velocities over a grassland
(0.3 ± 0.2 mm s$^{-1}$), suggesting eddy covariance as the main deposition driver. In a recent study, Jacobi et
al. (2019) confirmed the previous estimates suggesting that approximately 60% of the BC particles are
deposited on the surface snow via wet deposition in spring in the Svalbard Arctic area. Models are still
not fully able to describe the actual deposition and transport processes in Svalbard, resulting in
underestimating the BC concentration in the snowpack (Eckhardt, S. et al 2015, Stohl, A. et al. 2013).
Although wet deposition is suggested to be the main driver of BC concentration in the snow, little is
known about other environmental processes potentially affecting the BC particles concentration once
deposited, i.e. physical post-depositional processes.
In this study we present two unique experiments performed in a clean area close to the town of
Ny-Ålesund (Svalbard) at the Gruvebadet Aerosol Laboratory (78.91734 N, 11.89535 E, 40 m a.s.l.),
during spring 2014 and 2015. Daily and hourly time resolution samplings were performed on the snow
surface to investigate which atmospheric variables could directly or indirectly modify the BC mass
concentration in the surface snow once deposited. The daily sampling lasted for approximately 85 days to
assess the seasonal variability covering the transition from a cold period (April) to the melting period in



late May. The hourly time resolution experiment was performed to investigate potential processes
affecting the BC concentration over the diurnal cycle.

**2. Experimental Methods**
**2.1 Study Area**
Both experiments were conducted in the proximity of the Ny-Ålesund research station (78.5526
N, 11.5519 E, 25 m a.s.l.), located on the Spitzbergen island in Svalbard archipelago. Along the west
coast, Svalbard is characterized by a maritime climate with an annual average temperature of -3.9°C in
Ny-Ålesund (between 1994 and 2017) (Maturilli et al., 2019). On average, the snowpack starts building
up in September and melts away at the end of May (Førland et al. 2011). Ny-Ålesund has become one of
the reference locations for conducting Arctic climate studies focusing on atmospheric composition and
physics. Long-term monitoring of atmospheric aerosols is performed at the Gruvebadet station (Feltracco
et al., 2019; Moroni et al., 2018; Ferrero et al., 2016; Bazzano et al., 2015; Moroni et al., 2015;
Zangrando et al., 2013; Scalabrin et al., 2012), and at the Zeppelin observatory (475 m a.s.l.)
(Eleftheriadis et al., 2009; Tunved et al., 2013; Lupi et al., 2016, and reference therein).

**2.2 Snow Sampling**
There are no standardized methods for sampling, filtering and analytical protocols for detecting
atmospheric carbon deposited in snow, even if a few protocols have been developed (Ingersoll et al.,
2009; Gallet et al., 2018; Meinander et al., 2020). In the present work two different sampling strategies
were adopted regarding the thickness of the sampled layer and the temporal sampling frequency.
Snow samples were collected during two field campaigns: The first campaign was carried out in
Spring 2014, from April 1st to June 24th for a total of 85 days, it consists of daily sampling and it is
referred hereafter as the "85-days experiment". The second campaign was conducted in Spring 2015 from
April 28th to May 1st. During these three days, measurements were collected with hourly sampling. This
second campaign is hereafter referred as the "3-days experiment". Snow samples were collected about 1
km North-West of Ny-Ålesund (Figure 1). The area is a dedicated clean site for aerosols and snow
sampling, with no fuel engine traffic. The wind at the site is usually blowing from east to west, and rarely
from North to South, minimizing the emission of the town reaching the sampling area. The main wind
pattern during the experiment is presented in Figures 1 and 2. The samples for both experiments were
kept frozen until the lab analyses. The samples were collected using neck nylon gloves to avoid any
contamination.
The two experiments aim to capture the rBC mass concentration on a daily basis in the surface
snow (upper 10 cm) during the seasonal change and on an hourly basis on a thinner surface snow layer


(upper 3 cm) during a daily cycle. Although wet and dry deposition are the main sources of BC in the
Artic snow, the aims of our experiments were to evaluate if other atmospheric parameters could
contribute to the snow surface rBC mass concentration variability.

In the 85-days experiment, the first 10 cm of surface snow were collected on a daily basis
(approximately at 11.00 am, GMT+2) in the same area, using a 5 cm diameter and 10 cm long Teflon
tube. The samples were collected following a straight line leaving about 15 cm between the sampling
points to minimize the spatial variability. The collected snow was homogenized in a pre-cleaned plastic
bag and then, without melting, 50 mL was transferred into vial (Falcon™ 50mL Conical Centrifuge
Tubes) for BC, coarse mode particles number concentration and electrical conductivity analyses. The 85-
days experiment was designed with the aim to investigate the BC presence in the upper snow layer, where
most of the snow-radiation interaction takes place and where BC particles' presence can decrease the
snow albedo (Doherty et al., 2010). Moreover, this sampling strategy allowed to evaluate the variation of
BC on a seasonal basis and to capture the impacts of wind, precipitation or melting.

During the 3-days experiment, the first 3 cm of surface snow were collected on an hourly basis in
pre-cleaned vials in a delimited area of  2  x 2 m using the same sampling tools as above (Spolaor et al.,
2019). In this case the samples were collected following a straight line leaving about 5 cm between the
sampling points. The aim of the 3-days experiment was to investigate the potential daily cycle of surface
BC concentration; therefore, we foresaw that small variations could derive from the impact of the daily
variation of solar zenith angle and subsequent induced snow metamorphism at the surface of the
snowpack, often at cm scale. To avoid dilution of the signal, we reduced the vertical sampling thickness
to 3 cm to enhance our chances of observing variation in the rBC mass concentration if such variation
exists.

The temperature at the surface of the snowpack (at 7 cm for 85-days and at 3 cm for 3-days
experiment) was always measured. The daily/hourly snow accumulation was determined by measuring
the emerging part of 4 poles placed around the sampling area. The average standard deviation calculated
from the four poles provides us a reasonable estimate of the variability in snow accumulation\depletion
within the sampling area. The standard deviation obtained ranges from 2 to 4 cm for the entire periods,
indicating a limited spatial variability.

**2.3 Atmospheric Optical Measurements**
**2.3.1 Aethalometer (AE-31)**

In this study, the equivalent BC (eBC) concentration in the Boundary Layer (around 3 m a.s.l)
was measured by an AE-31 aethalometer (Gundel et al., 1983), during the 3-day campaign. The device is
equipped with 7-wavelengths (370, 470, 520, 590, 660, 880, 950 nm). It determines the attenuation





coefficient by using the light attenuation ratio through a sensing spot and a referenced clean spot, both on
a quartz fiber filter substrate. The sampling and reference spots surface areas are 0.5 cm$^2$, while the
volumetric flow rate is 4 l min$^{-1}$. The flow rate was calibrated with a TetraCal (BGI Instruments)
volumetric airflow before and after the field campaign. A 5 minutes temporal resolution was used for data
acquisition. However, due to the low background concentration in the Arctic, the signal/noise ratio is
high, so that data were hourly averaged. The data presented in this study were processed according to
Segura et al. (2014) methodology. For this purpose the multiple scattering and filter loading effect
(Weingartner et al., 2003) was corrected with new values of mass absorption cross section (MAC) and
multiple scattering factor (C=3.1), reported by Zanatta et al. (2018). The MAC value was derived using
observations and observationally constrained Mie calculations in spring at the Zeppelin Arctic station
(Svalbard, 78°N). Zanatta et al. (2018) estimated the MAC at 550 nm (9.8 m$^2$ g$^{-1}$) and at 880 nm (6.95 m$^2$
g$^{-1}$), which we used to estimate MAC at 520 nm (10.2 m$^2$ g$^{-1}$).

**2.3.2 Particle Soot Absorption Photometer (PSAP)**

During the 85-days sampling period the aerosol absorption coefficient was also measured by
means of a 3-wavelengths PSAP (this instrument was not available during the 3-days experiment period).
It measures the variation of light transmission through a filter where particles are continuously deposited
with constant airflow. A second filter identical to the first one remains clean and is used as a reference to
take into account possible variations of the light source, i.e. a 3-color LED (blue, green and red with
wavelength centred around 470, 530 and 670 nm, respectively). The correction developed by Bond et al.
(1999) was applied to consider the filter loading effect. The complete eBC mass concentration time series
for the 85-days experiment was retrieved using the Aethalometer (first period) and the PSAP (second
period), with an overlapping period with simultaneous measurements of 5 days. For the retrieved eBC
mass concentration from the two instruments to be equal during the overlapping period, the PSAP eBC
was calculated dividing the absorption measurements (at 530 nm) with a MAC equal to 7.25 m$^2$ g$^{-1}$
(keeping the AE31 data as reference). Daily averages were calculated from the 1-minute data to compare
with the rBC daily data obtained from the snow.

**2.4 Surface Snow measurements**

**2.4.1 Coarse Mode Particles Number Concentration**

The snow samples were melted at room temperature before the on-line coarse-mode particles and
conductivity measurements (the water was pumped from the vials by a 12 channels peristaltic pump,
ISMATECH, type ISM942). Specifically, the number concentration of coarse mode particles in the
surface snow was measured with a Klotz Abakus laser sensor particle counter. This instrument optically


counts the total number of particles and measures each particle's size in a liquid constantly flowing
through a laser beam cavity (LDS 23/23). The measurements size range of the instrument is from 0.8 to
about 80 µm with 32 dimensional bins (Table SI 1), not overlapping with that of the SP2. Only the 32nd
bin has a dimensional range above 15.5 µm, i.e. of 80 µm. The data were recorded by a LabView® based
software obtaining a sufficient number of data points in order to have a standard deviation of the mean
smaller than 5%. The particles number concentration was calculated using the constant water flow value.

**2.4.2 rBC Measurement – SP2**
The rBC mass concentration and mass size distribution were measured following the methods
described in Lim et al. (2014). The snow samples were melted at room temperature prior to the analyses.
The vials with the melted snow were sonicated for ten minutes at room temperature. The samples were
nebulized before the injection in the Apex-Q desolvation system (APEX-Q, Elemental Scientific Inc.,
Omaha, USA). The nebulization efficiency was evaluated daily by injecting Aquadag® solutions with
different mass concentrations, ranging from 0.1 to 100 ng $g^{-1}$, obtaining an average value of 61%, that
was used to correct all the BC mass concentrations reported in this manuscript. More details on the
method can be found in Lim et al. (2014) and in Wendl et al. (2014).
The SP2 data were analyzed using the IGOR based toolkit from M. Gysel (Laboratory of
Atmospheric Chemistry, Paul Scherrer Institute, Switzerland). The large amount of signals derived from
every single particle are elaborated achieving rBC mass and number concentrations and size distributions.

**2.5 Meteorological Parameters**
Meteorological parameters, in addition to the atmospheric and snow ancillary measurements,
were used in the statistical exercise to study the variability of rBC mass concentration in surface snow
samples as a function of the atmospheric conditions. BC particles are deposited on the snowpack
following a combination of wet and dry deposition. However, once deposited on/in the snowpack other
processes can potentially induced a significant variability in the surface BC content. The wind direction
and its velocity can modify the BC distribution in the upper snowpack due to snow-mobilization. The
solar radiation and relative humidity may enhance snow sublimation and surface hoar formation thus
modifying the relative BC concentration in the upper snow layer by removing or adding "water" mass to
the snow surface.
Air temperature and relative humidity at 2 meter height have been retrieved from a meteorological station
located about 800 meters north of the sampling site, using a ventilated PT-100 thermo-couple by Thies
Clima and a HMT337 humicap sensor by Vaisala, respectively. Wind speed and direction at 10 meter
height were obtained from a Combined Wind Sensor Classic by Thies Clima (see Maturilli et al., 2013).





At about 50 m distance, the radiation measurements for the Baseline Surface Radiatio Network (BSRN)
provide among others the downward solar radiation detected by a Kipp&Zonen CMP22 pyranometer
(Maturilli et al., 2015). Both meteorological and surface radiation measurements are available in a 1-
minute time resolution via the PANGAEA data repository (Maturilli et al., 2020). The daily/hourly mean
values of the meteorological parameters were used in the statistical analyses of the 85-days/3-days
experiment and in Figure 2 and Figure 3 (the physico-chemical parameters from the snow samples are
punctual values).

**2.6 Parameters consider in the statistical analysis**
The snow pack evolution is primarily driven by meteorological parameters, which are responsible for
adding/removing mass to the annual snow pack. Wind can affect the snow pack evolution in several
ways: 1) by snow redistribution, 2) favouring the ablation\sublimation, and 3) lifting particles from
nearby sources and areas. Surface snow and air temperatures are two fundamental parameters required to
fully understand the varying conditions of the snow pack. In our study, the temperature variables are
proxies for the melting episodes and for the presence of liquid water potentially affecting the
concentration of impurities. The incoming solar radiation is not expected to be directly linked to the
surface mass concentration of rBC, however the surface process could affect it indirectly by favouring
sublimation (water mass removal), as well as hoar formation (water mass addition) during the colder parts
of the day (night/early morning). The relative humidity gives an idea of the amount of water present in the
atmosphere and the high RH might favour the deposition of BC suspended by the formation of water
droplets through the cloud condensation nuclei, this is especially significant for the selected sampling
location, nearby to the shore. The last meteorological parameter considered is the precipitation amount.
This is important to understand the wet deposition processes able to transfer BC particles from the
atmosphere to the snow surface.
The additional selected parameters are 1) the atmospheric eBC mass concentration, an interesting
parameter to investigate the potential transfer function of BC particles from the atmosphere into the snow
surface, 2) the coarse mode particles (dust) that could have a similar transport pathways to the black
carbon and gives an idea of the amount of total impurities deposition and 3) the total water conductivity,
an indirect measurement of the salinity content of the snow. Considering the location of the sampling site
(<1km from the coast line), the contribution of the ocean emissions to the snow pack chemical
composition is significant. We considered the total conductivity as an indication of sea spray deposition,
and to investigate common deposition pattern and/or similarities to the behaviour of BC (although BC is
not emitted from ocean surface). The conductivity was also considered to determine if there was a large



sea-spray aerosol event, which could bring a lot of salt, potentially affecting the SP2 measurements (see
supplementary material).

**2.7 Statistical Analysis**
Multiple linear regression was carried out to evaluate the relationship between the observed
surface snow rBC mass concentration and the selected set of covariates consisting of the meteorological
and snow physico-chemical parameters that could have a direct effect controlling snowpack dynamics as
well on the BC concentration as discussed in Section 2.6. All the atmospheric parameters described in the
previous section (wind, snow and air temperature, incoming solar radiation, relative humidity and snow
precipitation amount) were initially considered as covariates to be included in the multiple linear
regression. However, wind speed and direction, as well as the atmospheric stability, expressed as vertical
wind speed, were removed because preliminary statistical analyses indicate that none of them is
associated with the observed variations in snow rBC mass concentrations. This does not mean that such
parameters do not play a role in controlling the BC concentration, but that no statistically significant
associations were found with the data collected in our study and thus these parameters no longer
considered in the statistical analyses discussed below.
Hence, the fitted multiple regression models are designed to describe the variation in snow rBC
concentrations as a function of atmospheric eBC concentration, surface snow coarse mode particles
number concentration, snow temperature (7 cm depth for 85-days experiment and 2 cm depth for the 3-
days experiment), snow precipitation, solar radiation and conductivity. Since the covariates considered in
the two experiments have quite different unit scales, the covariates have been standardized before fitting
the regression models. The standardization simplifies the comparison among the estimated effects of the
different covariates and between the two experiments, in this way facilitating the interpretation of the
results and their discussion. Further details about the statistical analyses are given in the Supplementary
material.
It is important to note that the eBC and the rBC mass concentrations are not the same physical
quantities: the former is obtained from an absorption measurement assuming a constant MAC, whereas
the second is obtained via a laser-induced-incandescence method with an SP2 empirically calibrated with
a reference material (Petzold et al., 2013).

**3. Results and Discussions**
**3.1 Seasonal BC variability in surface snow**
**3.1.1 Atmospheric eBC concentrations**



During the experimental period, the atmospheric eBC concentration shows a noticeable variability ranging from 80 ng m$^{-3}$ to < 5 ng m$^{-3}$ (Figure 2). The highest concentrations were measured at the beginning of the campaign, especially from April 15 to 27, followed by a general decreasing trend characterized by the presence of several concentration peaks (on May 8, 17 and 24) potentially due to Eurasian fires, as already suggested from Feltracco et al., 2020 (Figure S1). The ammonia daily concentration time series (the only available biomass burning tracer for that period in the area) measured at the Zeppelin station is plotted together with the Gruvebadet atmospheric BC measurements in Figure S3. Biomass burning is a significant source of atmospheric ammonia (Andreae and Merlet, 2001), often affecting the Arctic region (Moroni et al. 2020). As shown in Figure S3, both time series have a similar behaviour at the very beginning of the campaign, from April 3 to 8 and during the period between May 7 and 21. This suggests that the BC detected in the atmosphere could be originated from biomass burning episodes during these two time periods.

### 3.1.2 Surface Snow and Atmospheric Conditions

During the 85-days sampling period, wind was characterized by the following median values (25$^{th}$ and 75$^{th}$ percentiles) for direction and speed: 205° (152°, 257°) and 2.7 (1.9, 3.7) m s$^{-1}$, respectively, therefore mostly coming from South-West (Figure 2). Daily air temperature at 3 m increased during the campaign from -15°C to about +5°C (Figure 2) following the seasonal variation of incoming solar energy: from 100 to 300 W m$^{-2}$ with an average of 185 ± 75 W m$^{-2}$ (Figure 2, orange line). The snow precipitation episodes are presented as daily-accumulated values (Figure 2, blue bars) ranging from zero to 12 cm. The atmospheric eBC mass concentration, derived from the PSAP absorption coefficient, shows a decreasing trend during the campaign, ranging from 2 to 80 ng m$^{-3}$ with an average of 34 ± 23 ng m$^{-3}$.

Over the 85 days experiment, the snow rBC mass concentration varies from 0.2 to 6 ng g$^{-1}$ (Figure 2), with an average of 1.4 ± 1.3 ng g$^{-1}$, in agreement with results available in the literature (Mori et al., 2019; Jacobi et al., 2019; Aamaas et al., 2011). An increasing trend can be observed for the rBC mass concentration in the surface snow across the sampling period. The median of the rBC mass equivalent diameter in the snow is 313 ± 35 nm (Figure 2), similar to what obtained in other studies (e.g. Schwarz et al., 2013). The rBC mass equivalent diameter show high variability, ranging from 200 to 500 nm. However, since the rBC concentrations were low, the evaluation of the particles geometric mean diameter for the biggest sizes, above 300 to 400 nm, has only to be considered as qualitative information given the high noise in the size distributions.

The number concentration of coarse mode particles (Figure 2, blue line) shows a constant concentration in the first half of the campaign, until May 11, whereas increases in the second half, especially after the 1$^{st}$



of June, in concomitance with the onset of the snow melting period; the average number concentration is
$4914 \pm 4109$ # ml$^{-1}$. The conductivity (Figure 2, green line) shows as well an increasing trend at the end
of the sampling campaign when snow is melting, with an overall average value of $30 \pm 8$ µS. The spatial
variability of BC, calculated in the same manner as proposed by Spolaor et al. (2019) for other species,
was obtained from 6 surface snow samples collected in the four corners of the sampling area and two in
the centre right before the beginning of the experiment. The following rBC mass concentrations were
obtained: a) 3.95 ng g$^{-1}$; b) 4.92 ng g$^{-1}$ c) 4.20 ng g$^{-1}$d) 3.10 ng g$^{-1}$e) 3.82 ng g-1 f) 3.58 ngg$^{-1}$, resulting in
a BC spatial variability of 16% in the surface snow in the considered sampling area.

**3.1.3 Statistical Results**
The fitted multiple linear regression model for the 85-days experiment data explains 69% of the total
variance of snow rBC mass concentration ($R^2 = 0.69$). Statistically significant associations are found
among the snow rBC mass concentration and the coarse-mode particles number concentration ($p <$
0.001), the amount of snow precipitations ($p = 0.03$) and the snow temperature ($p < 0.001$). The statistical
associations of rBC mass concentration with the other covariates considered in the model were non-
significant (see Table 1 reporting the standardized estimated coefficients and the corresponding p-values).
Figure 4 displays both the 95% confidence intervals for the 85-days campaign and the 3-days experiment
in a way to allow a visual comparison of the estimated statistical associations between the snow rBC mass
concentration and the considered parameters.
In order to interpret the statistical results, the description of the 85-days campaign is split into two periods
identified as the transition from the "cold" to the "melting" state. The first period occurred before the end
of May: the rBC mass concentration often increases with snowfall episodes (April 9/10/11 and 17; May
17, 22 and 27/28; June 1) as suggested by previous studies, with exceptions for April 24 and May 7. Over
the sampling period, a weakly statistically significant positive association ($p = 0.03$) was found between
snow rBC mass concentration in surface snow and the daily amount of snow precipitation. BC wet
deposition processes are estimated to remove 50% - 60% of the total atmospheric BC burden in the Arctic
(Liu et al., 2011; Jacobi et al., 2019). In our study, the wet deposition impacts could be partially masked
due to the sampling frequency and the wind snow. In Kongsfjord, a strong wind is often present when
precipitation events occurred (Figure 2). Consequently, the freshly deposited snow is frequently removed
from the surface before being able to sample it. Usually, the sampled snow at the surface is not made of
the freshly precipitating but by redistributed surface snow. Interestingly, our observations show that, on a
daily scale, the precipitation episodes are not clearly related to a decrease in the atmospheric eBC mass





concentration (Figure 2). A possible explanation is that the precipitation amounts were small so that the
precipitation events did not alter significantly the atmospheric BC reservoir.
In the second period, from the beginning of June, the atmospheric temperature increases, causing the
snow-melting season's onset. At the beginning of June, the snow rBC mass concentration increases up to
approximately 5 ng g$^{-1}$, and a simultaneous increase was detected in the coarse mode particles number
concentration (peaks between June 4 and 7). As suggested in previous studies, the surface melting process
could explain the observed increase in BC and dust particles concentrations. However, we also have to
consider that both BC and dust can be dry deposited. Dry deposition is the main depositional process for
the coarse mode particles. Recently it has been suggested to have a significant contribution to the BC
surface content (up to 50-60%; Liu et al., 2011; Jacobi et al., 2019). Very few field validation data exist
for estimating the amount of dry deposited at the snow surface, and this process is often used as an
ancillary information since most models underestimate the BC in the Arctic snowpack compared to field
measurements.
Our data support the hypothesis related to local sources' activation in enhancing the dry deposition
impacts in an old mining town as Ny-Alesund. Especially during poor snow cover conditions, as during
the snow-melting season, dust particles as residuals of carbon extraction mining activities are available
for wind lift\suspension. The possible effect of local sources' activation is further supported by a recent
analysis of the Broggerbreen glacier and Ny-Alesund annual snow pack. This analysis shows the presence
of Retene (an organic compound frequently used to track the presence of coal), most likely due to local
sources (Vecchiato et al., 2018).
The simultaneous increase of rBC mass and coarse mode particle number concentrations during the
second part of the experiment (e.g. visible between June 3 and 7-8) could be explained via similar post-
depositional processes: snow melting and sublimation. The episodes of snow surface melting can
significantly affect the snow particulate content and we hypothesize that the hydrophobicity of pure BC
particles, and of several species in the coarse mode particles, might affect its physical location in the
snowpack (in the literature, the response of the BC particles is still debated): the hydrophobicity of the
particles can cause the surface concentration to increase while losing water mass through percolation.
This could lead into a positive feedback process: the increase of BC concentration can thus enhance snow
sublimation (water evaporation) resulting in a further increase of BC concentration in surface snow, and
so on.
In this study, the estimated statistical association between snow rBC mass concentration and the daily
snow temperature is negative and strongly significant (p < 0.001). During the 85-days experiment, we can





distinguish two events where the temperature appeared to play a role in the BC concentration. Both of
them show an increase in rBC mass concentration during melting/refreezing episodes, in agreement with
other studies (Aamaas et al., 2011). The first event occurred between May 5 to 12 and the second after
May 20, when the proper snow melting began (Figure 2). The first event was characterized by a rapid rise
of daily air temperature (from -6°C to -1°C) in concomitance to a snow precipitation event, followed by a
rapid temperature decrease to -6 °C. The surface snow (10 cm) mirrored this behaviour, first rising from -
6 °C to 0°C, and then cooling down to -6 °C. During this warm event, the upper snow strata underwent a
melting episode with surface water percolation (although limited), making the surface BC concentration
to increase. The second event started approximately on May 20 and lasted until the end of the experiment
(Figure 2). During this period, the atmospheric temperature increased constantly, and the snowpack
started to melt consequently. Moreover, surface BC concentration increased almost continuously from
May 25 to its maximum observed in June 6. Afterward, the upper snow rBC mass concentration tended to
decrease following the rapid snowpack decline.

### 3.2 Diurnal variation of rBC in surface snow

### 3.2.1 Surface Snow/Atmospheric Aerosol Content and Atmospheric Conditions

The 3-days experiment was performed at the end of April 2015, during the Arctic spring. The
samples were collected on an hourly basis over 3 days achieving a high-resolution sampling frequency.
The atmospheric concentration of eBC ranged from 2 to 50 ng m$^{-3}$, decreasing during the sampling period
and not showing any particular diurnal pattern (Figure 3). The mean value of the atmospheric eBC mass
concentration is $34 \pm 23$ ng m$^{-3}$, similar to the average of the 85-days experiment.
The surface snow rBC mass concentration exhibited hourly variability showing up to 2-fold
hourly increases (especially during the first day), overlapping a daily variation (Figure 3, bottom panel,
smoothed dark blue line). rBC mass concentrations of approximately 15 ng g$^{-1}$ were measured in the snow
samples from the beginning of the sampling to the end of the second day. Later, from the beginning of the
third day until the end of experiment, rBC mass concentrations show an average concentration of about 5
ng g$^{-1}$ (Figure 3). The average value over the whole sampling period is $9.5 \pm 5.2$ ng g$^{-1}$ (approximately 6
times higher than during the 85-days experiment). The rBC mass size distribution was characterized by a
median value of the geometric means of about $230 \pm 32$ nm, significantly lower than that which was
measured during the 85-days, and still in agreement with previous studies (Sinha et al., 2018; Bond et al.,
2013). The concentrations of EC and OC measured in parallel snow samples (not of the same volume) are
reported and described in Figure S4; the interpretation of the differences between the rBC and the EC
measurements in snow samples was beyond this manuscript's objectives.



The number concentration of coarse mode particles remains stable in the first half of the
experiment, until the end of April, and shows an average value over the three days of $26642 \pm 9261$ # ml⁻
³. The water conductivity shows a similar behaviour, and it is characterized by an average of $39 \pm 9$ µS
(30% higher than during the 85-days experiment).
All the measured snow impurities time series show two common features: first, a decrease in the
absolute values detected between 4 and 8 a.m. of April 30, despite the absence of precipitations and of
any particular meteorological episode (Figure 3); second, the impact of the snow precipitation event from
approximately 4 p.m. to midnight of the April 30, where the concentrations of aerosols in the snow
slightly increased at the very beginning whereas decreasing at the end of the event. Only the BC core
diameter remained above the average when the other aerosol snow content decreased (up to
approximately 400 nm of mass equivalent diameter), consequently returning to the average value.  The
spatial variability of BC, calculated as proposed by Spolaor et al. (2019) for other species, was obtained
by the analysis of 5 surface snow samples, collected in the four corners of the sampling area and one in
the centre obtaining the following concentrations: a) 10.17 ngg⁻¹, b) 10.64 ngg⁻¹, c) 7.04 ngg⁻¹, d) 11.98
ngg⁻¹, and e) 11.91 ngg⁻¹, thus resulting in a spatial variability of 19%. Clear sky conditions where
observed for the duration of the sampling period except for the snowfall occurred at the end of the third
day.

**3.2.2 Statistical Results**
The multiple linear regression model for the 3-days experiment explains the 83% of the total snow rBC
mass concentration variance, a percentage higher than the 85-days experiment, likely due to the more
stable atmospheric conditions and the greater interaction with the atmosphere of the upper 3 cm compared
to the 10 cm used for the seasonal experiment. The fitted multiple linear regression model indicates a
statistically significant association between the rBC mass concentration in the snow and the conductivity
($p < 0.001$), the number concentration of coarse-mode particles ($p = 0.003$), the snow precipitation
amount ($p < 0.001$), the incoming solar radiation ($p = 0.009$) and the snow temperature ($p = 0.01$). The
standardized estimated coefficients are reported in Table 1, displayed along with 90% and 95%
confidence intervals in Figure 4.
The association between the coarse-mode particles number concentration and the snow rBC mass
concentration is positive and strongly significant ($p < 0.001$), similarly to what observed for the 85-days
experiment, confirming the similar behaviour of these types of particles.
A negative association is found between the rBC mass concentration in the snow and the
incoming solar radiation ($p = 0.009$), and a weaker negative association with the snow temperature ($p =$
$0.01$). The latter is strongly dependent on the solar radiation. This relation suggests that the rBC mass



concentration in surface snow might undergoes to a diurnal variation: low mass concentrations when the
solar radiation is high and vice versa. The BC particles are known to be non-volatile and not photo-
chemically active, therefore the decrease in their concentration observed when the solar radiation is
higher could not be explained as a re-emission process from the snowpack into the atmosphere as
observed for other aerosol species (Spolaor et al., 2018; Spolaor et al., 2019). The results show that the
highest rBC mass concentration levels are detected in the samples collected in the late afternoon. The late
night/early morning concentration decrease is connected with the surface hoar formation (clear sky
condition is essential for the hoar formation) able to dilute the surface snow BC concentration.
Specifically, the lowest rBC mass concentration value is found between 5 a.m. and 12 a.m. and in the
same time interval the solar radiation increases from 100 to 400 W m$^{-2}$, followed by a delay of the air and
the snow temperatures increase. In these time frames, the temperature offset between the air and the
surface snow is the highest, up to 4°C, with the surface snow being the coldest between the two.
Condensation of water vapour on the top of the snow crystals is likely adding "water" mass (without BC
particles) in the collected samples and diluting the original rBC mass concentration. This process could
also explain the positive statistical association between snow rBC mass concentration and conductivity ($p$
$< 0.001$, Table 1 and Figure 4) mostly influenced by the presence of sea salt in the snow samples. In fact,
considering the proximity of the sampling site to the coastline ($< 1$ km), the marine spray deposition
mainly controls the total conductibility. The slight increase in conductivity, as well as in the sodium
concentration (Spolaor et al., 2019), determined during the night time could be associated to the formation
of ice nuclei from the sea spray aerosol particles present in the atmosphere surrounding snowpack. The
lower night temperature could exponentially increase the ice nuclei formation, favouring the deposition of
suspended sea spray aerosol (DeMott et al., 2016).

The snow precipitation amount is negatively associated with the rBC mass concentration in the

snow ($p < 0.001$). As previously remarked, the aerosol scavenging intensity is not measurable with snow
sampling strategies based on the sampling of a constant snow thickness from the surface (3 cm in this
case). We tentatively explain the negative relation observed in this study with the high frequency
sampling, being able to follow the evolution of the BC particles scavenged during a snow episode (from 3
to 12 p.m. of the 30[th] April 2015). The beginning of the precipitation episodes appeared to remove the
highest amount of BC particles, leaving the atmosphere cleaner as reflected by the lower BC mass
concentration revealed in subsequent samples. The snow collected at 18:00 of April 30 showed a higher
amount of rBC as well as the highest coarse mode particles number concentration and conductivity. In the
next few hours, from 9 to 12 p.m., the snow precipitations were depleted in terms of aerosol content and
rBC mass concentration.



From the 3-days experiment, it appeared that the physical processes affecting the surface of the
snowpack, like surface hoar formation and sublimation, play an important role. Therefore, the physical
characteristics of the snow layers in which BC is embedded should be more studied in order to better
characterize the daily variations of BC and its impact on the albedo. The 3 days experiment took place
under clear sky conditions (most of the time) and Arctic like atmospheric circulation (Figure S2): this is
crucial for investigating the variations observed, highlighting the impacts of the parameters following a
diurnal cycle (ISR, snow metamorphism). This daily variability showed that the highest concentration of
rBC is found during mid-day/afternoon, when the incoming solar radiation is high. In conclusion, the
combination of snow metamorphism, which normally occur during a daily cycle, associated to the
observed variability of rBC surface mass concentration could slightly modify the snow albedo during a
day cycle. Although this effect might have a minor impact, more detailed studies including snow density
and optical snow grain radius measurements should be pursued at centimetre vertical resolution in order
to correctly estimate the radiative impact of the daily rBC variations, and confirm the findings from the
proposed experiment.

**4. Conclusions and Future Perspectives**
The seasonal and daily experiments (85- and 3-days long, respectively) suggest that the rBC
concentration in the upper snow layer is not only due to a cumulative process such as when evaluating the
entire annual snow pack but, rather by a more complex process involving atmospheric, meteorological
and snowpack conditions. Our results based on a multiple linear regression models suggest that the
amount of BC in the surface snow is decoupled from the BC atmospheric load. This finding suggests that,
despite the potentially high atmospheric BC concentrations (as in the case of long-range transport of
biomass burning plumes), the surface snow BC mass concentration can potentially remain unaffected. In
both experiments the coarse-mode particles are positively associated to the snow BC mass concentration,
suggesting that the BC and dust deposition undergo similar deposition and post-depositional processes in
the upper snowpack. Specifically, before the beginning of the melting season, the wet deposition episodes
appeared to have major impacts, whereas the activation of common local sources favour the wind
resuspension from uncovered areas enhancing the intensity of dry deposition processes, triggering an
accelerated snow melting positive feedback.
Our results also suggest that in order to explain the observed BC mass concentration variability
during seasonal and diurnal time ranges other processes than wet and dry depositions should be
considered. Post-depositional processes, as snow sublimation and melting, can remarkably affect the rBC
mass concentration. Sublimation and hoar formation are affecting the BC content in the uppermost thin
layer by adding/removing water mass, thus explaining the observed BC diurnal cycle (3-days hourly



sampling experiment). On the other hand, the surface melting episodes enrich the BC content in the surface layer not because of enhanced deposition but mainly because of water mass loss. In particular, the snow mass loss is stronger during the snow-melting season, where an increase in the rBC concentration could significantly alter the snow albedo and further enhance the radiative absorption, hence promoting a positive feedback. We believe our results to be representative at least of the Arctic costal areas, characterized by similar processes and seasonality.

The remarkable diurnal and daily variability, as well as the complex interdependent mechanisms affecting the rBC mass concentration in the Arctic surface snow, makes the results of albedo-based radiative impact model of the active layer a potential source of erroneous conclusions: the impacts of long distance biomass burning episodes might be overestimated, whereas the impact of local sources and dry deposited impurities during the melting season underestimated. Further empirical studies are therefore necessary in order to improve our understanding of the involved physical mechanisms and to better constrain modelling studies.

**Acknowledgements**

This work was part of the PhD (in "Science and Management of Climate Change") of Michele Bertò at the Ca' Foscari University of Venice that was partly funded with the Early Human Impact ERC project. Thanks to Giuseppe Pellegrino for helping collecting the samples. Thanks to Jacopo Gabrieli and the technicians of the Ca'Foscari University of Venice for the precious help in building up the coarse mode particles and conductivity measurement apparatus. We acknowledge the use of data and imagery from LANCE FIRMS operated by the NASA/GSFC/Earth Science Data and Information System (ESDIS) with funding provided by NASA/HQ. We want to thank Paolo Laj and the LGGE (Grenoble, France) for lending us the SP2 and Marco Zanatta for transferring the SP2 know-how on instrumental functioning and data analyses. Thanks to Martin Gysel-Beer, PSI, for the IGOR based SP2 Toolkit for SP2 data analyses. We thank Marion Maturilli and AWI for providing us with the meteorological data. Thanks to Giorgio Bertò for checking and correcting the language of this manuscript. This paper is an output of the AMIS project in the framework of "Project MIUR – Dipartimenti di Eccellenza 2018-2022". This project has received funding from the European Union's Horizon 2020 research and innovation programme under grant agreement No 689443 via project iCUPE (Integrative and Comprehensive Understanding on Polar Environments).





**Data Availability**

Meterological and surface radiation data are available at the PANGAEA database (Maturilli, 2015a;
2015b; 2015c; 2016a; 2016b; 2018a; 2018b; 2018c; 2018d; 2018e). The data for precipitation amount at
Ny-Ålesund can be accessed via the eKlima database of MET Norway. The BC data are available upon
request.

**Author Contributions**
Author contributions. AS, EB, DC and MB conceived the experiments; AS, EB, DC, and LP collected the
samples; MB measured the samples; KM and MMaz provided the atmospheric eBC concentrations; SC
and DC provided the back-trajectories analyses; CV performed the statistical analyses with inputs from
MB and AS. MB prepared the manuscript mainly with inputs from AS, J-C. G and DC (in the methods
section from AS, KM, MMaz) and all co-authors contributed to the interpretation of the results as well as
manuscript review and editing.

**Data repository**
Maturilli, Marion (2020): Basic and other measurements of radiation and continuous meteorological
observations at station Ny-Ålesund  (April, May 2014 and April, May, June 2015), reference list of 10
datasets. Alfred Wegener Institute - Research Unit Potsdam, PANGAEA,
https://doi.pangaea.de/10.1594/PANGAEA.913988 (DOI registration in progress)















**FIGURES**
**Figure 1.** a) Experimental sampling site location (dark grey rectangle), in proximity of the Gruvebadet
Aerosol Laboratory. b) Gruvebadet area (black square), close to the Ny-Ålesund research village. From:
Spolaor et al., 2019 (maps from https://toposvalbard.npolar.no/). The red arrow points to the North.

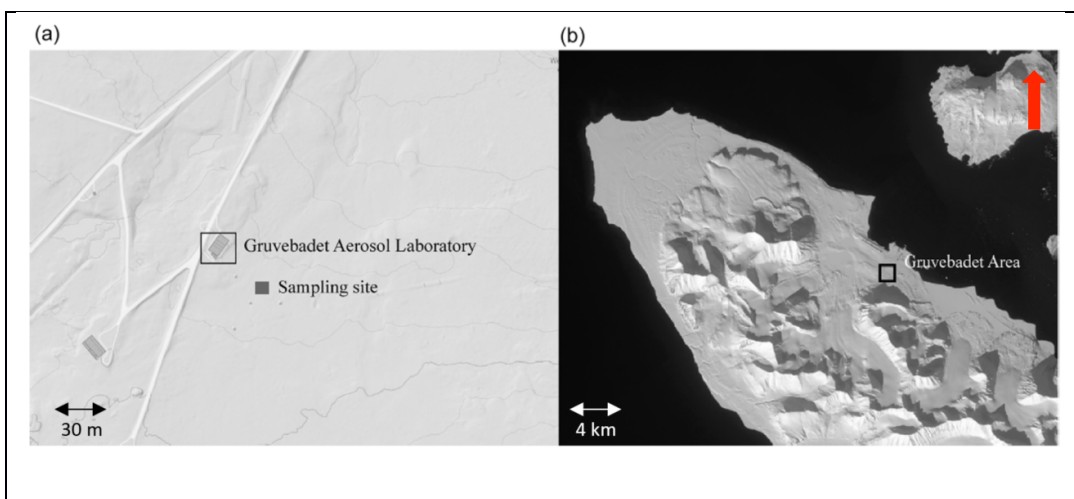

















**Figure 2.** The 85-days experiments daily snow samples rBC mass concentration (light blue), eBC mass
concentration in the atmosphere (black), geometric mean mass equivalent diameter (purple), number of
coarse mode particles (blue), total conductivity (green), meteo/snow parameters used in the statistical
exercise: wind speed color coded for wind direction, solar radiation (orange line), air and surface snow
temperatures (blue bars and green line respectively), amount of fresh snow ("snow precipitations", light
blue bars) and the snow accumulation ("Neg. accumulation"; the values where multiplied by -1 in order to
show the similar trend of the snow lost and of the air/snow temperature during the melting period at the
end of the campaign).

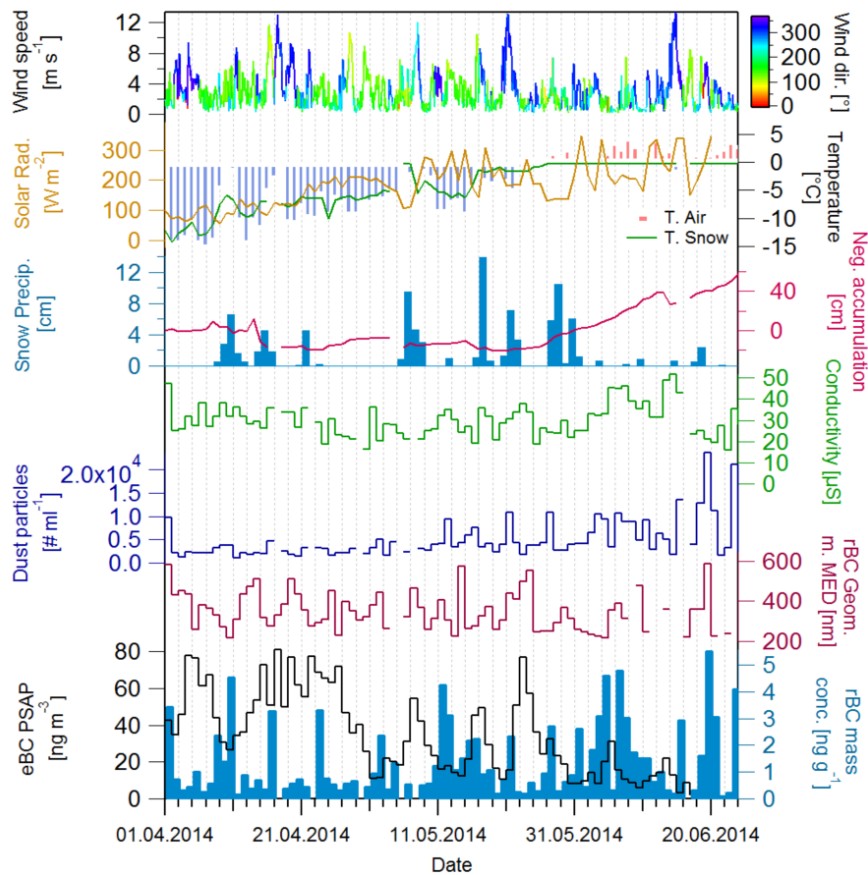





**Figure 3.** The 3-days experiments snow samples hourly rBC mass concentration and smoothed line (light
blue bars), atmospheric eBC mass concentration in the atmosphere (black), geometric mean mass
equivalent diameter (purple), the number concentration of coarse mode particles (blue) and the total
conductivity (green), meteo/snow parameters used in the statistical exercise: wind speed color coded for
wind direction, solar radiation (Orange line), Air and surface snow temperature (blue bars and green line
respectively), amount of fresh snow ("snow precipitations", light blue bars). The yellow bars are centered
on the midnight hours for the three sampling days.

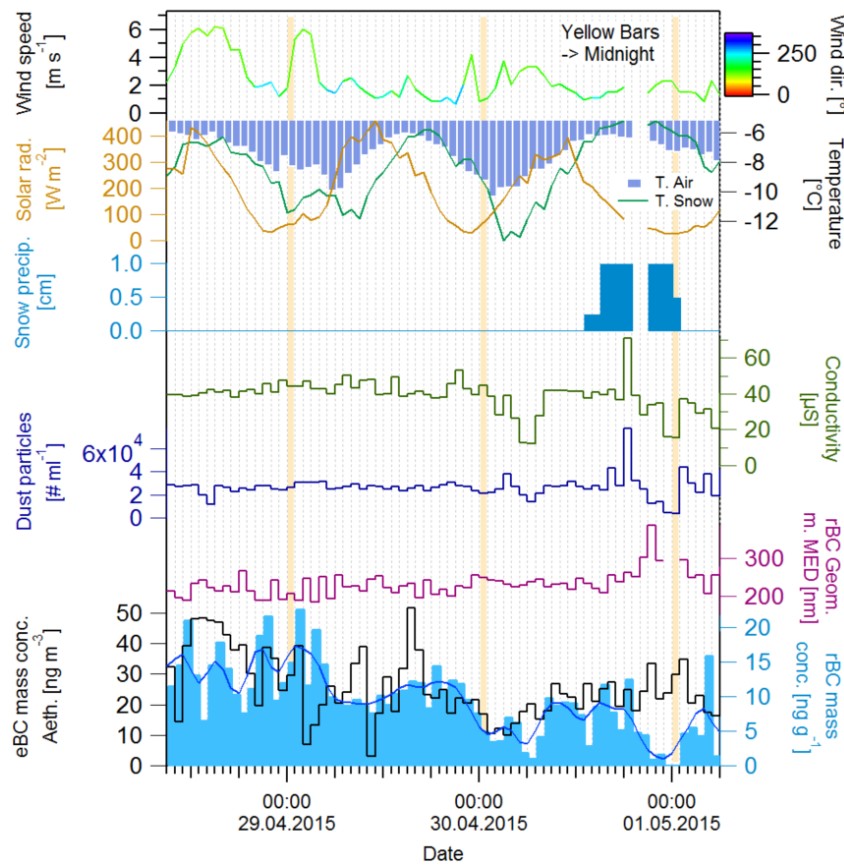









**Figure 4.** Standardized estimated coefficients of the multiple linear regression models fitted to the 3 days
and 85 days experiments. The segments correspond to 95% confidence intervals about the corresponding
estimates. The internal thicker segments correspond to 90% confidence intervals. Intervals that do not
include the zero correspond to statistically significant covariates. If a confidence interval consists of
positive values, then there is a significant positive association between the corresponding covariate and
snow rBC mass concentration given the remaining covariates. Vice versa, if the confidence interval
consists of negative values, then the association is negative. The abbreviations used in the plot are:
"log(cond)" – logarithm of the water conductivity time series, "log(dust)" – logarithm of the coarse mode
particles number concentration time series, "eBC" – equivalent black carbon atmospheric concentration,
"snow" – amount of fresh snow from the precipitation episodes, "SWR" – solar radiation, "temp" – the
snow temperature. The plot is produced with the R package (R Core Team, 2020) jtools (Long, 2020).

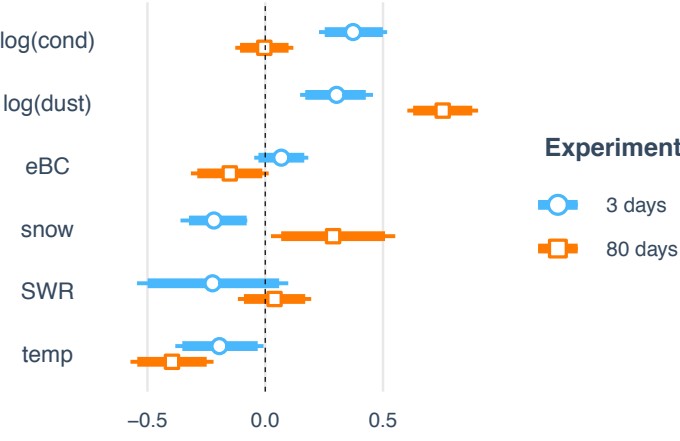












**TABLES**
**Table 1.** Standardized estimated coefficients and p-values for the multiple linear regression models fitted
to the 3 days and 85 days experiments data. The intercept and the trigonometric terms used to account for
the 24-hours periodicity in the 3-days experiment are not displayed.

| Covariate | 3 days | 85 days |
|---|---|---|
| **log(cond)** | 0.38 ($p < 0.001$) | -0.00 ($p = 0.95$) |
| **log(dust)** | 0.23 ($p = 0.003$) | 0.75 ($p < 0.001$) |
| **eBC** | 0.06 ($p = 0.26$) | -0.15 ($p = 0.07$) |
| **snow** | -1.02 ($p < 0.001$) | 0.29 ($p = 0.03$) |
| **SWR** | -0.43 ($p = 0.009$) | 0.04 ($p = 0.61$) |
| **temp** | -0.23 ($p = 0.01$) | -0.40 ($p < 0.001$) |
| **$R^2$** | 0.83 | 0.69 |
















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
