# Peer review of "Variability of Black Carbon mass concentration in surface snow at Svalbard"

_Atmospheric Chemistry and Physics, 2021_

## Author Response (AR1)

**Reply to Anonymous Referee #1**

**While this paper was submitted as a new paper, it is a revision of a paper previously submitted. I am reviewing it as I would a revised paper since I also reviewed the original version. The paper is significantly improved from the earlier version. It still suffers from some of the difficulties of the original, but I will recommend publication with revision.**

We thank the referee for having appreciated our efforts to improve the manuscript. We also thank the referee for her\his work in revising again the manuscript with instructive comments and suggestions. Please find below our point-by-point reply.

**As noted in my review of the original paper, the difficulty is that the sampling and analysis wasn't really designed to answer the questions being posed by this analysis. (Most specifically the snow sampling was done to a fixed depth in both experiments, rather than being designed to isolate, e.g., the influence of newly fallen snow, melt layers, hoar frost, etc.) As noted by the other reviewer of the original paper, correlative analysis is also not really the right tool to be using here for processes-level insight (e.g. see comment 1 below), and that is still the approach taken. There isn't anything that can be done at this point about the sampling, and the more focused analysis presented in this revised paper is an improvement. The data collected a useful contribution to the quantification of snow BC concentrations in the Arctic so with correction of the issues below I recommend publication.**

We are convinced that the sampling strategy we propose is correct for the aim of our study since we are focusing on the variability in the BC concentrations in the upper snow layer (10 cm) where these impurities can have the higher impact in the albedo reduction\change. This aim is now clearly stated in the new title of the manuscript. We cannot state that our approach is the best but we believe it is the appropriate solution for obtaining a general overview of the processes that could affect the BC concentration in the surface snow pack. We fully agree with the reviewer that if we had investigated the effect of deposition, then we would have needed to sample only the fresh snowfall at consequently different depth resolutions. However, such a study is not the goal of our project.

Our statistical analysis is not just a study of the pairwise correlations between the available parameters. Indeed, we consider a multiple linear regression model to account for the *joint effect* of different meteorological and physicochemical parameters on the BC concentrations in the surface snow pack. We acknowledge that the previous version of the paper lacked the necessary details to understand our statistical analyses. Section 2.7 of the revised paper now provides more details about our statistical modelling approach. Moreover, following the reviewer's criticism about the presence of a diurnal cycle, we revised the statistical analysis of the three-days experiment as discussed later in this point-by-point reply.

**1) There is still some difficulty with the explanations for the selected variables. See my comments below about SZA as a driving variable for snow rBC concentrations. The reasoning behind expecting surface snow temperature to correlate with daily variations in surface snow rBC concentrations is unclear. Once melting commences, any time the temperature goes above freezing surface rBC will continue to increase. Based on physical understanding of process driving surface snow concentration changes you don't expect there to be a correlation between air temperature and surface snow concentration because the process of surface snow BC concentrations increasing with melt is cumulative; when temperatures go back down, there's no physical reason why rBC would then decrease. Therefore applying correlation at the hourly or daily timescale doesn't make sense.**

We corrected the refuse about SZA that, indeed, does not affect the rBC surface snow mass concentration. In the revised paper we replace SZA with SWI.

The snow temperature is a fundamental parameter necessary to understand the physical process of the snowpack, of the upper snow layer and the meteorological condition. We understand the point of the reviewer, in particular during the 85-days experiment. When the snow melting began, the snow temperature stayed around zero and presented no significant oscillations suggesting that the change in the BC is dependent on other variables. However, the snow temperature could be relevant during the entire experiment indicating the possible snow metamorphism, the response of the upper snowpack to meteorological conditions, including spring warming events (T > O°C), and the begin of the snowpack melting. The snow temperature is not driving the rBC during the melting phase, but the statistical analysis should be equally carried out during the entire experiment and not only during the "cold" phase of the experiment. Instead in the case of the 3-days experiment, the snow temperature can be used as an indication of the response of the upper snowpack to the change induced by the SWR. Correlation between rBC with SWR and snow temp could indicate that process with a daily scale frequency, could affect the upper snow rBC concentration. Essentially the temperature in the 3 days is a tracer of the daily oscillation. Therefore, we have improved the description in the Section 2.6 of the revised paper (line 260-266).

**2) The one major issue with the paper that still needs to be addressed is that it argues there is a diurnal cycle in rBC observed during the 3-days experiment (lines 420-421 and 465, referencing Figure 3), then hypothesizes that the formation and evaporation of hoar frost is driving this cycle. However:**

**i.) Most problematic of all is that I don't see the purported diurnal cycle in rBC that is referenced as existing in Figure 3. As noted in my review of the original paper, the dark blue line in the bottom panel simply looks like smoothed random variations, superimposed on an overall decreasing trend. If the authors are going to argue there was a diurnal cycle and then explain why this diurnal cycle exists, they first need to show that there actually IS a diurnal cycle with some sort of statistical analysis. Simply asserting it's there is not sufficient.**

**ii) No observations are presented to support that hoar frost formed as hypothesized. Was hoar frost observed during the 3-days experiment measurements? Were the weather conditions (e.g. clear skies and low winds) conducive to hoar frost formation?**

Thanks for the sensible comments. Our previous discussion on the rBC diurnal cycle was based on the statistical analyses. However, as observed by the reviewer, the diurnal cycle is not that evident. Indeed, the statistical significance of the diurnal cycle was somehow weak and based on a relatively short experiment (three days). Accordingly, we revised the statistical analyses finding that a multiple regression model without the diurnal cycle (i.e. removing the two trigonometric terms that described the diurnal cycle in the statistical model) gives a more robust summary of the 3-days experiment data. The revised statistical results indicate no significant association between rBC and SWR or temp, that is the two parameters that undergo to the diurnal cycle. Accordingly, we completely rewrite Section 3.2.2 of the revised paper on the basis of the new statistical findings.

**Smaller comments:**

**3) Line 149: Solar zenith angle is cited here as a likely primary driver of snow BC concentrations. It's really solar insolation that is relevant, and then really only as it drives temperature and therefore snow metamorphism changes (e.g. melting and hoar frost formation). Indeed, "Solar radiation" (W/m2) is what's shown in Figures 2 and 3, along with temperature. This text should be edited to reflect this.**

Thanks for pointing out this wrong sentence; the revised paper has been modified accordingly (line 154).

**4) lines 253-255 then 256-258: I don't understand this explanation given that RH is an observational variable of interest since "high RH might favour the deposition of BC suspended by the formation of water droplets through the cloud condensation nuclei." It's then stated that precipitation amount is monitored, so that would be a more direct measure of wet deposition (if that's indeed the type of deposition monitoring RH was supposed to reveal).**

RH is a common meteorological parameter. High values of RH are not always associated to wet precipitation (for example, foggy conditions occurring in the Arctic during Spring). However, high values of RH could favour the formation and the grown of atmospheric water droplet in this way favouring the dry deposition. The amount of precipitations, in terms of water mass, caused by this process is negligible compared to a proper snowfall, but the grown of the water droplet around the cloud condensation nuclei can favour the deposition of the rBC in the surface snowpack. There is no direct link between RH and rBC but might an indirect relationship since it can be used to better understand the meteorological condition. For example, low RH conditions; clear sky and absence of cloud cover (for several days) can favour the sublimation of the snow from the surface. RH, similar to temperature, is not a primary driver of rBC deposition but it could promote\favour the dry deposition. This experiment is unique and it has the aim to explore possible environmental parameters able to influence the rBC concentration in the upper snow pack. We revised the lines 272-275 of the paper to make this point clearer.

**5) Lines 301-302 vs lines 320-322: The same result is expressed two different ways in two different places. Why? Lines 301-302 express this as "variability", then notes it declined through the campaign; lines 320-322 simply expresses it as a trend. The partitioning of information into Sections 3.1.1 (atmospheric eBC) and Section 3.1.2 (atmospheric conditions) is a bit odd.**

Thanks for pointing out this issue. The sentence at line 301-302 has been modified from "During the experimental period, the atmospheric eBC concentration shows a noticeable variability ranging from 80 ng m$^{-3}$ to < 5 ng m$^{-3}$ (Figure 2)" to "During the experimental period, the atmospheric eBC concentration range between from 80 ng m$^{-3}$ to < 5 ng m$^{-3}$ (Figure 2) with an average of 34 ± 23 ng m$^{-3}$" and we remove the sentence at line 302-322 since is just a repetition. We also re arrange the section 3.1.1. and 3.1.2. as suggested.

**6) "coarse mode particles" (used in text) and "dust particles" (Figures 2 and 3) are used interchangeably. What's really shown in the figure is coarse mode particles. In much literature "dust" has a fairly specific meaning (mineral dust). Here, the coarse mode particle composition isn't determined so it would be best to stick with terminology that reflects what is actually measured. In the text you could then note that the coarse mode particles are likely a mix of soil, mineral dust and (as noted) possibly coal dust.**

We agree with the referee and we modified accordingly the corresponding text in the revised paper. In both figures we we use the acronymous Cmp (coarse mode particles).

**7) Lines 308-311: The number concentration of coarse mode particles in snow, as noted, is lower during the first half of the 85-days campaign, then increases during the second half of the campaign (Figure 2). On line 355 an 'average concentration' of 4914+/-4109 per ml is given but it's not clear whether this is across the full duration of the campaign (in which case it's not a very meaningful number, given the clear difference between the first and second half of the campaign) or it's only over the second half of the campaign (in which case the time period used for this statistic should be given).**

We improved the corresponding section in the revised paper as suggested (lines 352-356).

**8) lines 337 and 341: BC should be rBC**

Thanks for pointing out the error, now corrected in the revised paper.

**9) lines 373-375: "Dry deposition is the main depositional process for the coarse mode particles. Recently it has been suggested to have a significant contribution to the BC surface content (up to 50-60%; Liu et al., 2011; Jacobi et al., 2019)." This needs to be explained or written differently: Atmospheric BC is not a coarse-mode particle so dry deposition of coarse-mode particles as a significant source of BC to surface snow is a confusing statement.**

We agree with the referee and thus we rewrote the sentence in the revised paper to make it clearer (lines 391-394).

**10) Lines 379-382: "Our data support the hypothesis related to local sources' activation in enhancing the dry deposition impacts in an old mining town as Ny-Alesund. Especially during poor snow cover conditions, as during the snow-melting season, dust particles as residuals of carbon extraction mining activities are available for wind lift\suspension." It's argued that this is a significant source of BC to snow. Yet on lines 536-537 it's stated that: "We believe our results to be representative at least of the Arctic coastal areas, characterized by similar processes and seasonality." Svalbard is fairly unusual for the Arctic in the degree to which past coal mining is likely influencing the addition of rBC to the snowpack from local surface sources (lines 379-358), so how can you assert that you think the results here are generalizable to the Arctic coastal areas overall (lines 536-537)?**

Thanks for the reasonable comment. The Ny-Alesund area is ideal to perform snow research studies since the large amount of monitoring programs on-going and the supporting datasets available. However, there are some limitations at Ny-Alesund derived from the fact that it was a coalmine till the 60s. At the beginning and at the end of the snow season, the rBC concentration in the surface snow can be influenced by the characteristics of the experiment site. During the winter and spring periods the snow cover is homogeneous (with the notable exception of extraordinary dry years) and thus the upper snowpack is mainly driven by the atmospheric deposition. We improved the conclusions of the revised paper as suggested by the reviewer (see lines 399 and 544-548).

**11) lines 393-395: The impact of surface snow melt on surface snow concentrations of particulates is presented too much as a hypothesized process; in fact there is significant support for this in the literatures. In addition to the modeling that simulates this and in addition to Aamaas et al 2011 there are at least three other studies showing this in observational data:**

**Xu, B., T. Yao, X. Liu, and N. Wang, Elemental and organic carbon measurements with a two-step heatinggas chromatography system in snow samples from the Tibetan Plateau, Ann. Glaciol., 43, 257–262, doi: 10.3189/172756406781812122, 2006.**

**Doherty, S. J., T. C. Grenfell, S. Forsström, D. L. Hegg, S. G. Warren and R. Brandt, Observed vertical redistribution of black carbon and other light-absorbing particles in melting snow, J. Geophys. Res., 118(11), 5553-5569, doi:10.1002/jgrd.50235, 2013.**

**Doherty, S. J., D. A. Hegg, P. K. Quinn, J. E. Johnson, J. P. Schwarz, C. Dang and S. G. Warren, Causes of variability in light absorption by particles in snow at sites in Idaho and Utah, J. Geophys. Res. - Atmos., 121, doi:10.1002/2015JD024375, 2016.**

Thanks: we added the suggest references\papers at line 420.

**12) lines 424-425: It's noted that the concentration of rBC in snow is 6x higher in the 3-days experiment than in the 85-days experiment. Earlier it's pointed out that the concentrations during the 85-days experiment were consistent with those found in previous studies. Any thoughts on why the very large increase in snow rBC?**

We can only propose two hypotheses to explain the differences in the rBC concentrations  (see the lines below) and this is why we do not investigate this aspect in details in the paper.

One explanation regards the different sampling depths: In the 85 days experiment we sampled the upper 10 cm where the rBC can be more diluted in the snow mass collected compared to the 3 days experiment. This suggest that the rBC tends to accumulate in the upper layer, but this conclusion (???) is strongly dependent on the meteorological conditions. Higher accumulation of rBC in the upper layer requires a relative long period of absence of snow fall and strong wind that favours the dry deposition and the rBC accumulation in the upper snowpack. However, we cannot prove this argument since the 3 days experiment were designed for another scope.

The second possible explanation is the interannual variability of rBC for the selcted site and the influence of a specific atmospheric deposition event before the 3-days experiment. The surface snow samples collect during the 3-days experiment could be affected by a single deposition event able to increase the rBC concentration. However, link a specific atmospheric event (reconstructed with the back-trajectory approach) to the explanation of the hourly surface snow rBC variability in Ny-Alesund is rather speculative mostly because of the orography around the experimental site. The samples collected during the 3-days experiment are, most likely, not representative for the site but just a snapshot during the 3 days of the experiment.

**13) line 427: Bond et al. 2013 didn't give snow rBC sized so this isn't an appropriate citation. The work of Schwartz et al. could be cited instead.**

Thanks: we modify the revised paper accordingly (line 445).

**14) line 435: "All the measured snow impurities time series show two common features…: Which variables are your referring to here? The description that follows matches that for rBC but not for e.g. dust/coarse mode particle concentration. What do you mean by "all the measured snow impurities time series"?**

We Thanks: we added the reference to the supplementary material: Figure S4 and Section 4.

**15) line 520: Remaining "unaffected" is different from "not being a primary driver of variations in surface snow rBC over XX timescales". What you've shown supports the latter statement but not the former, and only sort of, since the observational period did not include any highly elevated atmospheric eBC periods.**

We agree with the referee and modified the revised paper accordingly (line 531).

**Reply to Anonymous Referee #2**

**The manuscript 'Variability of Black Carbon mass concentration in surface snow at Svalbard' by Michele Bertò et al. reports the cumulative processes (such as atmospheric, snowpack and meteorological conditions) in governing the refractory Black Carbon (rBC) mass concentrations in the upper snow layers at Svalbard. The database (85 days in 2014, 1 Apr-24 Jun and 3-days in 2015, 28 Apr- 1 May) in this study is useful to the characterization of aerosol-cryosphere interaction over the Arctic. However, there are many weak elements and lack of clarity in several aspects. I recommend publication of this manuscript in the Atmospheric Chemistry and Physics after my comments have been addressed.**

We appreciate the overall positive recommendation expressed by the reviewer. We wish also to thank the reviewer for her/his instructive comments and suggestions. Please find below our point-by-point reply.

**The major concern is the main outcome of the study. Authors state that 'precipitation events were the main drivers of the BC variability (line-33)". However, the snow precipitation amount is negatively associated with the rBC mass concentration during 3-days experiment, as against the positive association during 85-days experiment. How do authors explain these contrasting effects of precipitation on BC concentration in snow? In line 353-367, authors try to connect atmospheric eBC concentration with the wet scavenging processes, which requires better investigation. Do authors want to highlight (quantify?) how effective the wash out processes in compared to various other factors considered in this study?**

Thanks for the comments. The main difference can be explained from the different sampling strategy. During the 85-days experiment we sample at daily time resolution collecting the upper 10 cm. Sampling at daily resolution means that we are observing\measure the resulting of 24 hours of deposition (and removal) without evaluate how the rBC concentration evolve during a snow deposition event. With this strategy we will observe the final effect of a snow deposition event. The 3-days experiment is designed differently with the aim to evaluate the change at hour resolution. If we consider a snow fall event that last for 10 hours most of rBC suspended in the atmosphere is deposited in the first hours. Thereafter, the subsequent snow deposition will result in a depleted concentration of rBC since the atmosphere becomes "cleaner". Furthermore, since we are sampling the upper 3 cm, we collect only the last snow deposit and not the entire snow accumulated during the event. With the daily resolution (the 85-days experiment) we evaluate the net effect of a snow fall while the 3-days experiment with hourly resolution allows us to monitor the evolution of rBC concentrations in the deposited snow.

During the 85-days experiment we determine a significant correlation between the occurrence of snow deposition and rBC. This suggest that the wet deposition contributes to the rBC transfer from the atmosphere to the snow although the occurrence of a snow fall event is not always associated with an increase of rBC in the snow surface. For example, the snow event occurred around the 7[th] of May (Figure 2) is not associated to an increase of rBC.

The difference between the two experiments could be also found in the natural variability associated to the type of atmospheric transport\snow events. If a snow event is associated to air mass transport from the Arctic ocean, then the amount of rBC will be relatively low compared to the snow concentration. Vice versa if a snow event is associated to a transport from mid latitudes (or Siberia during the fire season), then the rBC is expected to be higher.

The aim of the paper is not to estimate or quantify the wash out for rBC, as it will be not be possible following our strategy, but rather to investigate which variable could affect the daily surface concentration. Accordingly, in the paper we have not stated any intent to calculate the wash out but we only describe the possible role that wet deposition could have in modify the rBC in the upper 10 cm.

**The interpretation of the standardized estimated coefficients derived from multiple linear regression models amongst rBC (in snow) and water conductivity, coarse mode particles number concentration, equivalent black carbon atmospheric concentration, snow precipitation episodes, solar radiation and the snow temperature during 85- and 3-days experiments lacks coherent interpretation as well as proper explanation of the physical processes involved. The positive and negative associations seen in the case of eBC, fresh snow and SWR during 85 days and 3-days experiments must be clearly described.**

Following the comments of the reviewer, we carefully revised:

1. the discussion of the selected variable in Section 2.6 of the revised paper;
2. the statistical model description in Section 2.7 of the revised paper;
3. the interpretation of the estimated model coefficients in Sections 3.1.3 and 3.2.2 of the revised paper.

We think part of the apparent incoherent results were mostly due to the presence of the trigonometric terms in the regression model for the 3-days experiment. These terms were included in the model to capture the diurnal cycle. However, the statistical significance of the diurnal cycle is somehow weak and based on a relatively short experiment (three days). Accordingly, we revised the statistical analyses finding that a multiple regression model without the diurnal cycle gives a more robust summary of the 3-days experiment data. The revised statistical results indicate no significant association between rBC and SWR or temp, that is the two parameters that undergo to the diurnal cycle. Accordingly, we completely rewrite Section 3.2.2 of the revised paper on the basis of the new statistical findings. We also revised the interpretation of the remaining parameters in order to make the presentation of the statistical results clearer.

**Information about both dry and wet deposition processes are randomly put in different context, this could be avoided**.

The discussion about the wet and dry depositions is repeated in the paper with the precise aim to explain specific condition of rBC change during the two experiments. Wet and dry depositions are the main drivers of the transfer of eBC from the atmosphere to the snow surface and these two processes should be carefully considered at any time.

**The BC in the upper snowpack affects the snow Albedo. There are multi-layer approaches to understand the effect of vertical distribution of BC in the snowpack (e.g., Dang et al., 2017). In this study, what is the criterion of selecting "upper snowpack" as 10 cm in 85-days and 3-cm in 3-days experiments?**

The two experiments were intentionally designed to describe the evolution of the rBC mass concentration in the surface snow in Svalbard. During the 85-days experiment we focused on the seasonal variability and we strategically adopted a daily resolution sampling frequency of the snow layer that are controlling the albedo (the uppermost 10 cm of the snowpack), at the same time limiting the rapid oscillation\change that could occur in the upper layer (3 cm) has seen for the the 3-days experiment. The light reflection and transmission occur mainly in the upper centimetre of the snow pack (line 146). On the other side, the 3-days evolution experiment was focused on the changes connected to daily variations induced by light and temperature; in order to maximize the measured impacts, we sampled only the first 3 cm of snow. Sampling the upper 3 cm for the 80-days experiment would have exposed the samples to oscillations we determined during the 3-days experiment due to physical snow processes (sublimation, hoar formation, etc.). On the contrary, by sampling the upper 10 cm we wanted to smooth out these effects while being still able to evaluate the changes occurring in the active surface snow. For instance, sampling 20 cm of snow would have cause a too drastic smoothing of the surface signal since the lower and more stable snow layers would have mainly driven the resulting concentration variability.

**If SZA is the primary driver of the diurnal variation of BC in snow, what about the cloud cover? SZA is important while estimating snow albedo change, but does it really influence (under the given circumstances) the BC concentration on a diurnal scale in 3 cm snow? How magnificent is snow metamorphism when the diurnal temperature remains below -6 C?**

Thanks for the comment: here the usage of SZA was a refuse that we amend in the revised paper as explained also to the other reviewer. We fully agreed that SZA cannot influence the snow metamorphism during the 3-days experiment. The snow metamorphism is not only dependent on the temperature but also on the gradient of the snow pack, maximised at the upper 20 cm of the snow pack. We use "snow metamorphism" not only to refer to the melting episode but also to all the other physical processes (sublimation\hoar formation) driven by temperature variations. We also underline that the discussion on the diurnal rBC diurnal cycle was fully based on the statistical results. However, as also suggested by the reviewer 1, the diurnal cycle is not that evident. In fact, the statistical significance of the diurnal cycle was weak and based on a relatively short series(3-days). Accordingly, we carefully revised the statistical model removing the diurnal cycle. The new results obtained from the revised statistical analysis indicate no significant correlation between rBC and SWR or temperature, the two parameters that undergo to the diurnal cycle. Accordingly, we completely rewrote Section 3.2.2. with the revised statistical findinds.

**Line 35-36: "The statistical analysis suggests that the BC content in the snow is decoupled from the atmospheric BC load." This is not clear.**

Thanks: we modify the text to make it clearer.

**Line 82-85: The citation about the dry deposition parameters is not suitable in the context of present study**

We remove the sentence "Emerson et al. (2018) empirically evaluated the in situ rBC deposition velocities over a grassland ($0.3 \pm 0.2$ mm s$^{-1}$), suggesting eddy covariance as the main deposition driver".

**Line 98: seasonal > intra-seasonal variability**

Thanks for point-out this typo: the text has been modified accordingly.

**Line 121/ 99: Sampling period is contradicting; is it May end or till June 24?**

Thanks for point-out this typo: the text has been modified accordingly.

**Section 2.3.1: In general, filter loading effect is negligible at Arctic due to very low BC concentrations. Why MAC at 530 nm is used? Do aethalometer derived absorption coefficients agree well with PSAP measurements? Why MAC = 7.25 m$^2$g$^{-1}$ is considered for PSAP estimates of eBC? Virkkula (2010) is more appropriate for PSAP data correction.**

The AE-31 and the PSAP used in the present work operate at slightly different wavelengths (530 and 520 nm, respectively) which may have caused some confusion, although this is stated in the text. The instruments also use different filter materials. The absorption coefficient of the PSAP operated at the Gruvebadet lab have been tested in various campaigns with several different instruments (AE-33, AE-31 and PAX) and also during a yearly-long campaign with a MAAP and thus they could be considered reliable. The AE-31 eBC values have been carefully tested in a previous Arctic campaign (Markowicz et al. 2017). For the present work, we used the AE-31 as the reference for the eBC absolute value. The PSAP data are fully consistent with those of the AE-31, in the period of the operational overlap, using a MAC of 7.25 at 530 nm which is well in the range (7.5 ±1.2 , Bond and Bergstrom 2006) suggested at this wavelength (see also Backman et al. 2017).

References

Backman, J., Schmeisser, L., Virkkula, A., Ogren, J. A., Asmi, E., Starkweather, S., Sharma, S., Eleftheriadis, K., Uttal, T., Jefferson, A., Bergin, M., Makshtas, A., Tunved, P., and Fiebig, M.: On Aethalometer measurement uncertainties and an instrument correction factor for the Arctic, Atmos. Meas. Tech., 10, 5039–5062, https://doi.org/10.5194/amt-10-5039-2017, 2017.

Bond, T. C. and Bergstrom, R. W.: Light absorption by carbona- ceous particles: An investigative review, Aerosol Sci. Tech., 40, 27–67, https://doi.org/10.1080/02786820500421521, 2006.

K.M. Markowicz, C. Ritter, J. Lisok, P. Makuch, I.S. Stachlewska, D. Cappelletti, M. Mazzola, M.T. Chilinski, Vertical variability of aerosol single-scattering albedo and equivalent black carbon concentration based on in-situ and remote sensing techniques during the iAREA campaigns in Ny-Ålesund, Atmospheric Environment, 164, 2017, 431-447, https://doi.org/10.1016/j.atmosenv.2017.06.014.

**Section 2.4.2: This section is very important, which explains the estimates of rBC in snow. However, the methodology using SP2 (please expand) for rBC (in snow) estimation is not clear. The following statement needs to be refined:**

**"The nebulization efficiency was evaluated daily by injecting Aquadag® solutions with different mass concentrations, ranging from 0.1 to 100 ng g-1, obtaining an average value of 61%, that was used to correct all the BC mass concentrations reported in this manuscript."**

We implement the section 2.4.2 in particular at line 216 – 226. Additional information about the method are reported in the supplementary material.

**Line 345-347: Please include the values of $R^2$?**

The R2 indices for the models fitted on the two experiments indicate an oversall satisfactory fitting quality (R2=0.69 for the 85-days experiment; R2 =0.78 for the 3-days experiment). These values are reported in the paper and also in Table 1. Please notice that we fit a multiple linear regression thus we have a single R2 index for each model. Each parameter has an associated estimated coefficient in the regression model whose statistical significance is summarized with the corresponding p-value given in the paper and also in Table 1.

**Line 381: "…. as residuals of carbon extraction mining activities", add reference to this statement**

Thanks for the suggestion: we included the reference Vecchiato et al. 2018.

**Line 432: Why average coarse mode number concentrations are significantly higher in 3-days experiment ($\sim 26642 \pm 9261$ ml[-1]) than that in 85 days experiment ($\sim 4914 \pm 4109$ ml[-1])?**

**Fig-3: Why rBC concentration is relatively higher in 3 days experiment (> 10 ppb) as compared to the values in the corresponding period of 85 days experiment (< 6 ppb)?**

We can only propose two hypotheses to explain the differences in the rBC concentrations (see the lines below) and this is why we do not investigate this aspect in details in the paper.

One explanation regards the different sampling depths: In the 85 days experiment we sampled the upper 10 cm where the rBC can be more diluted in the snow mass collected compared to the 3 days experiment. This suggests that the rBC tends to accumulate in the upper layer, but this conclusion (???) is strongly dependent on the meteorological conditions. Higher accumulation of rBC in the upper layer

requires a relative long period of absence of snow fall and strong wind that favours the dry deposition and the rBC accumulation in the upper snowpack. However, we cannot prove this argument since the 3 days experiment was designed for another scope.

The second possible explanation is the interannual variability of rBC for the selcted site and the influence of a specific atmospheric deposition event before the 3-days experiment. The surface snow samples collect during the 3-days experiment could be affected by a single deposition event able to increase the rBC concentration. However, link a specific atmospheric event (reconstructed with the back-trajectory approach) to the explanation of the hourly surface snow rBC variability in Ny-Alesund is rather speculative mostly because of the orography around the experimental site. The samples collected during the 3-days experiment are, most likely, not representative for the site but just a snapshot during the 3 days of the experiment.

**Fig-4: 80 or 85 days? In the regression model, how the values of SWIR are considered? Diurnal average or normal incident condition?**

We modify and correct in 85-days. The SWR data  for the regression model are daily average during the 85-days experiment and hourly during the 3-days experiment**.**

**Lines 465-469: "… low mass concentrations when the solar radiation is high and vice versa. The BC particles are known to be non-volatile and not photo chemically active, therefore the decrease in their concentration observed when the solar radiation is higher could not be explained as a re-emission process from the snowpack into the atmosphere as observed for other aerosol species". The sentence is not clear. Please rewrite.**

Thanks for the comment: the text has been modified to make the sentence clearer. Please notice that the statistical analyses have been revised by removing the trigonometric terms describing diurnal cycle from the statistical model fitted to the three-days experiment.  See also the discussed made some lines before in this point-by-point reply.

---

## Referee Report (RR1)

Comments on **acp-2021-39**
**Variability of Black Carbon mass concentration in surface snow at Svalbard by Berto et al.**

I appreciate the authors providing me with detailed responses to my previous comments, which help me understand this paper in-depth. I recommend the publication of this manuscript after minor modifications.

1. The abstract of the paper needs revision.

   (a) The statement "Two different snow-sampling strategies were adopted during spring 2014 and 2015 …" is misleading, as the 3-days of data is not representative of the entire spring season of the year 2015.

   (b) Line 32-33: "Precipitation events were the main drivers of the BC variability", this contradicts with the results stated in lines 503-513.

   (c) Line 34-35: "(wind resuspension in specific Arctic areas where coal mines were present)" is not clear.

2. Please include in section-2: How water conductivity was measured/ estimated?

3. Figure-4 caption: "given the remaining covariates", the sentence is not complete. 95% and 90% confidence interval can be marked in the figure.

4. Table-1: what is intercept in the first row?

5. Line 493: and additional process > an additional process

---

## Author Response (AR2)

**Comments on acp-2021-39**
**Variability of Black Carbon mass concentration in surface snow at Svalbard by Berto et al.**
**I appreciate the authors providing me with detailed responses to my previous comments, which help me understand this paper in-depth. I recommend the publication of this manuscript after minor modifications.**

We thank the reviewer(s) for appreciating the change we made and for the useful suggestion received during the entire review process improving the overall quality of the paper.

**1. The abstract of the paper needs revision.**

**(a) The statement "Two different snow-sampling strategies were adopted during spring 2014 and 2015 …" is misleading, as the 3-days of data is not representative of the entire spring season of the year 2015.**

**(b) Line 32-33: "Precipitation events were the main drivers of the BC variability", this contradicts with the results stated in lines 503-513.**

**(c) Line 34-35: "(wind resuspension in specific Arctic areas where coal mines were present)" is not clear.**

All the change request from the reviewer have been considered and change made in the abstract.

**2. Please include in section-2: How water conductivity was measured/ estimated?**

A sentence has been included at line 201

**3. Figure-4 caption: "given the remaining covariates", the sentence is not complete. 95% and 90% confidence interval can be marked in the figure.**

Thanks for the comment. The expression "given the remaining covariates" is commonly used in Statistics to mean "conditionally to the remaining covariates" because regression models are defined in way to have a natural "conditional interpretation". Although the two expressions are equivalent in Statistics, we have replaced "given" with "conditionally to" in order to enhance the readability.

The 95% and 90% confidence intervals are marked in the figure using the segments' thickness: "The segments correspond to 95% confidence intervals about the corresponding estimated coefficients. The **internal thicker segments** correspond to 90% confidence intervals".

**4. Table-1: what is intercept in the first row?**

The intercept corresponds to the coefficient beta0 that is included in the regression model (see the formula in line 301 of the paper). The intercept term is not of direct interest but it is important for a correct statistical interpretation of the remaining terms of the regression model and it is also necessary for the validity of the R2 index.

**5. Line 493: and additional process > an additional process**

We modify accordingly